# Structure of human Na$_V$1.6 channel reveals Na$^+$ selectivity and pore blockade by 4,9-anhydro-tetrodotoxin

Yue Li [1,2,3,7], Tian Yuan[4,5,7], Bo Huang [6,7], Feng Zhou[6], Chao Peng[4,5], Xiaojing Li[1,3], Yunlong Qiu[1,3], Bei Yang[1], Yan Zhao [1,3] ✉, Zhuo Huang [4,5] ✉ & Daohua Jiang [2,3] ✉

The sodium channel Na$_V$1.6 is widely expressed in neurons of the central and peripheral nervous systems, which plays a critical role in regulating neuronal excitability. Dysfunction of Na$_V$1.6 has been linked to epileptic encephalopathy, intellectual disability and movement disorders. Here we present cryo-EM structures of human Na$_V$1.6/β1/β2 alone and complexed with a guanidinium neurotoxin 4,9-anhydro-tetrodotoxin (4,9-ah-TTX), revealing molecular mechanism of Na$_V$1.6 inhibition by the blocker. The apo-form structure reveals two potential Na$^+$ binding sites within the selectivity filter, suggesting a possible mechanism for Na$^+$ selectivity and conductance. In the 4,9-ah-TTX bound structure, 4,9-ah-TTX binds to a pocket similar to the tetrodotoxin (TTX) binding site, which occupies the Na$^+$ binding sites and completely blocks the channel. Molecular dynamics simulation results show that subtle conformational differences in the selectivity filter affect the affinity of TTX analogues. Taken together, our results provide important insights into Na$_V$1.6 structure, ion conductance, and inhibition.

Voltage-gated sodium (Na$_V$) channels mediate the generation and propagation of action potentials in excitable cells[1,2]. In humans, nine Na$_V$ channel subtypes (Na$_V$1.1–1.9) had been identified, which are involved in a broad range of physiological processes due to their tissue-specific distributions in various excitable tissues[3,4]. Subtype Na$_V$1.6, encoded by the gene *SCN8A*, is ubiquitously expressed in neurons of both the central nervous system (CNS) and the peripheral nervous system (PNS), especially enriched in the distal end of axon initial segment (AIS) and in the node of Ranvier of myelinated excitatory neurons. The Na$_V$1.6 channel is believed to play a primary role in the initiation and propagation of action potentials in those neurons by lowering the threshold voltage[5–11]. Emerging evidence suggests that Na$_V$1.6 is also expressed in some inhibitory interneurons and plays a role in establishing synaptic inhibition in the thalamic networks[12–14]. Compared with other Na$_V$ channel subtypes, Na$_V$1.6 possesses unique biophysical properties including activation at more hyperpolarized voltage, higher levels of persistent current and resurgent current, and higher frequency of repetitive neuronal firing in neurons such as cerebellar Purkinje cells[15–23]. These features make Na$_V$1.6 a critical and favorable mediator in regulating neuronal excitability in those neurons. Meanwhile, dozens of mutations in Na$_V$1.6 have been linked to human diseases, most of which exhibit gain-of-function phenotypes, increase neuronal excitability, and cause different types of epileptic encephalopathy[24–28]; whereas loss-of-function mutations are often

[1]National Laboratory of Biomacromolecules, CAS Center for Excellence in Biomacromolecules, Institute of Biophysics, Chinese Academy of Sciences, Beijing 100101, China. [2]Laboratory of Soft Matter Physics, Institute of Physics, Chinese Academy of Sciences, Beijing 100190, China. [3]University of Chinese Academy of Sciences, Beijing 100049, China. [4]State Key Laboratory of Natural and Biomimetic Drugs, Department of Molecular and Cellular Pharmacology, School of Pharmaceutical Sciences, Peking University Health Science Center, Beijing 100191, China. [5]IDG/McGovern Institute for Brain Research, Peking University, Beijing 100871, China. [6]Beijing StoneWise Technology Co Ltd., 15 Haidian Street, Haidian District, Beijing, China. [7]These authors contributed equally: Yue Li, Tian Yuan, Bo Huang. ✉ e-mail: zhaoy@ibp.ac.cn; huangz@hsc.pku.edu.cn; jiangdh@iphy.ac.cn

associated with later onset seizures, intellectual disability, isolated cognitive impairment and movement disorders[29–31]. Thus, Na$_V$1.6 is an important drug target; effective and subtype-selective therapeutics are eagerly awaited for the treatment of Na$_V$1.6-related epilepsy and other neurological diseases.

Eukaryotic Na$_V$ channels are composed of a pore-forming α subunit and auxiliary β subunits[32]. The four-domain α subunit exerts voltage sensing, gate opening, ion permeation, and inactivation[4,33]. Meanwhile, one or two β subunits bind to the α subunit to regulate Na$_V$ channel kinetics and trafficking. Among the four types of β subunits[34–37], β1 and β3 subunits non-covalently bind to the α subunit, while β2 and β4 subunits are covalently linked to the α subunit via a disulfide bond[32,38]. To date, high-resolution cryo-electron microscopy (cryo-EM) structures of seven mammalian Na$_V$ channels (Na$_V$1.1–1.5, Na$_V$1.7–1.8) have been reported[39–45]. Together with the resting-state[46], open-state[47], and multiple ligand-bound Na$_V$ channel structures[48–50], these structures revealed the general molecular mechanisms of voltage-sensing, electromechanically coupling, fast inactivation, sodium permeation, and ligand modulation. Among those Na$_V$ channel modulators, the guanidinium neurotoxin tetrodotoxin (TTX) has long been used as a useful tool to study Na$_V$ channels, which can potently inhibit Na$_V$1.1–1.4 and Na$_V$1.6–1.7 at nanomolar level (TTX-sensitive Na$_V$ channels), and less potently inhibit Na$_V$1.5, Na$_V$1.8, and Na$_V$1.9 at a micromolar concentration (TTX-insensitive Na$_V$ channels). The detailed binding mode of TTX had been revealed in the Na$_V$ channel-TTX complex structures[44,51]. Furthermore, two guanidinium neurotoxin derivatives, ST-2262 and ST-2530, were reported as potent and selective inhibitors for Na$_V$1.7, indicating that TTX analogs could potentially be developed as selective therapeutics[52,53]. Interestingly, 4,9-anhydro-tetrodotoxin (4,9-ah-TTX), a metabolite of TTX, has been reported to selectively block Na$_V$1.6 with a blocking efficacy of 40- to 160-fold higher than other TTX-sensitive Na$_V$ channels[54]. However, the structure of Na$_V$1.6 and how 4,9-ah-TTX blocks Na$_V$1.6 remain elusive.

In this work, we show a fully-functional shorter-form construct of human Na$_V$1.6 suitable for structural studies, and present cryo-EM structures of Na$_V$1.6/β1/β2 apo-form and in complex with 4,9-ah-TTX. Complemented with electrophysiological results and molecular dynamics (MD) simulations, our structures reveal Na$_V$1.6 structural features, sodium conductance, and pore-blockade by 4,9-ah-TTX.

## Results

### Construct optimization of Na$_V$1.6 for cryo-EM study
To conduct structural studies of Na$_V$1.6, human wide-type Na$_V$1.6 (named Na$_V$1.6$^{WT}$) was co-expressed with human β1 and β2 subunits in HEK293F cells and was purified similarly to previously reported Na$_V$ channels[41,44]. Although the amino acid sequence of Na$_V$1.6 is highly conserved with other Na$_V$ channel subtypes (e.g., 70% identity with Na$_V$1.7); however, the purified Na$_V$1.6$^{WT}$ sample exhibited poor quality and did not permit high-resolution structural analysis (Supplementary Fig. 1a, b). Construct optimization had been proven to be successful in improving the sample quality of Na$_V$1.7 and Na$_V$1.5[55,56], we therefore carried out construct screening of human Na$_V$1.6 by removing unstructured intracellular loops and C-terminus. We found that deletion of S478-G692 between D$_I$ and D$_{II}$ (Na$_V$1.6$^{ΔDI-DII}$), S1115-L1180 between D$_{II}$ and D$_{III}$ (Na$_V$1.6$^{ΔDII-DIII}$), or R1932-C1980 of the C-terminus (Na$_V$1.6$^{ΔCter}$) showed improved sample homogeneity based on the size-exclusion chromatography (SEC) profiles (Supplementary Fig. 1a). Strikingly, when we combined these modifications and deleted all of the three unstructured regions, it displayed a sharp mono-disperse SEC profile, which is much better than that of Na$_V$1.6$^{WT}$ and any of the single-deletion constructs (Fig. 1a, b, Supplementary Fig. 1a). We next examined the functional characteristics of the triple-deletion construct by whole-cell voltage-clamp recording of Na$_V$1.6-expressing HEK293T cells. The candidate construct exhibits typical voltage-dependent activation and inactivation (Fig. 1c). The resulting V$_{1/2}$

values of the voltage-dependence of activation and steady-state fast inactivation are −31.3 ± 0.3 mV ($n = 15$) and −77.3 ± 0.2 mV ($n = 15$), respectively, which are close to the reported V$_{1/2}$ values of human wide-type Na$_V$1.6[57,58]. These results confirmed that the triple-deletion construct fulfills similar electrophysiological functions to the Na$_V$1.6$^{WT}$. The preliminary cryo-EM analysis of this triple-deletion construct showed that the micrograph contains a rich distribution of mono-disperse particles, which gave rise to much better 2D class averages with well-resolved features than the Na$_V$1.6$^{WT}$ (Supplementary Fig. 1b, c). Thus, this triple-deletion construct (named Na$_V$1.6$^{EM}$) was selected for further structural studies.

### The overall structure of human Na$_V$1.6
The purified Na$_V$1.6$^{EM}$/β1/β2 sample was frozen in vitreous ice for cryo-EM data collection (Supplementary Fig. 2). After processing, the final reconstruction map from the best class of ~41 k particles was refined to an overall resolution of 3.4 Å (Fig. 2a, Supplementary Figs. 3–5). As expected, the resulting Na$_V$1.6$^{EM}$/β1/β2 structure closely resembles the reported structures of human Na$_V$ channels due to high sequence similarity (Fig. 2b). For example, the binding modes of the β subunits are consistent with the structures of human Na$_V$1.7/β1/β2 and Na$_V$1.3/β1/β2[41,44]; the pore-forming α-subunit of Na$_V$1.6$^{EM}$ can be well superimposed with Na$_V$1.7 with a backbone (1107 Cα) root mean square deviation (RMSD) of 1.4 Å (Fig. 2c). However, marked local conformational differences were observed between the two structures, especially in the extracellular loops (ECLs) (Fig. 2c, d). The ECLs are less conserved regions among the nine Na$_V$ channel subtypes (Supplementary Fig. 6a), which form the outer mouth of the selectivity filters (SFs) and contribute to the binding of β subunits. Superposition of the Domain I ECLs of Na$_V$1.6$^{EM}$ and Na$_V$1.7 shows that the ECL$_I$ of Na$_V$1.6$^{EM}$ lacks the short α2 helix, but instead forms an extended hairpin-like turn (Fig. 2d). Importantly, the ECL$_I$ of Na$_V$1.6$^{EM}$ exhibits more N-linked glycosylation modification sites than Na$_V$1.7; N308-linked glycosylation site appears to be unique for Na$_V$1.6 based on the sequence alignment (Supplementary Fig. 6a). Although these structural differences in the ECLs do not affect the binding of β subunits to Na$_V$1.6 (Fig. 2a), the glycosylation and other modifications shape the surface properties of Na$_V$1.6, which play important roles in its trafficking, localization, and pathology[59,60]. For instance, a unique glycosylation site in the ECL$_I$ of Na$_V$1.5 blocks the binding of the β1 subunit to Na$_V$1.5[43].

We next compared the fast inactivation gate and intracellular activation gate between Na$_V$1.6$^{EM}$ and Na$_V$1.7, which only display subtle conformational shifts (Fig. 2e, f), indicating that those key structural elements are highly conserved to fulfill their similar biological roles. Consistently, the signature fast inactivation gate, Ile-Phe-Met motif (IFM-motif), binds tightly to its receptor site adjacent to the intracellular activation gate (Fig. 2e), resulting in a non-conductive activation gate constricted by A411, L977, I1464, and I1765 from the four S6 helices, respectively (Fig. 2f). The van der Waals diameter of the activation gate is less than 6 Å, suggesting that the gate is functionally closed (Fig. 3a, b).

### Potential Na$^+$ sites in the SF
The ion path of Na$_V$1.6 has two constriction sites, the extracellular SF and intracellular activation gate, respectively (Fig. 3a, b). The sodium selectivity of mammalian Na$_V$ channels is determined by the extracellular SF[61,62], which is composed of an Asp from D$_I$, Glu from D$_{II}$, Lys from D$_{III}$, and Ala from D$_{IV}$, known as the DEKA-locus[63,64]. Based on structural analysis, the acidic residues of the DEKA-locus are believed to act as a high-field strength site, which attracts and coordinates Na$^+$; and the Lys in D$_{III}$ was proposed as a favorable binding ligand for Na$^+$ which facilitates the ions passing through the SF[43,65]. In coincidence with other mammalian Na$_V$ channels[43,44], the SF of Na$_V$1.6$^{EM}$ adopts an asymmetric conformation composed of the DEKA-locus (Fig. 3b, c). No oblivious Na$^+$ binding site had been identified in previous structures of

mammalian Na$_V$ channels. In contrast, densities for Ca$^{2+}$ were consistently reported in the structures of bacterial Ca$_V$Ab channel and mammalian Ca$_V$1.1, Ca$_V$2.2, and Ca$_V$3.1 channels[66–69]. Interestingly, two strong blobs of EM densities were observed in the SF of Na$_V$1.6$^{EM}$ (Fig. 3d, e), which are deduced as potential Na$^+$ binding sites because Na$^+$ ions are the only major cations in the solutions throughout the purification processes. The upper site (namely Na1) closely engages E936 of the DEKA-locus and an additional acidic residue E939 (Fig. 3c). The distances of this Na1 to the E936 and E939 are at ~3.5 Å, suggesting that Na$^+$ in Na1 site may still be hydrated. Meanwhile, D370 of the DEKA-locus contributes minorly to this Na$^+$ binding site at a distance of ~7.5 Å (Fig. 3c). This observation is in line with previous studies showing that E936/K1413 of the DEKA-locus are the most prominent residues for Na$^+$ permeation and selectivity, while D370 of the DEKA-locus is not absolutely required[63]. This potential Na1 site may represent the first step for Na$^+$ conductance, that is, E936 of the DEKA-locus attracts and captures one hydrated Na$^+$ from the extracellular solution with the assistance of E939. The second blob of density is located inside the SF, namely the Na2 site, which is about ~5.3 Å away from the Na1 site (Fig. 3d, e). Interestingly, the Na2 is close to the short side-chain residue A1705 of the DEKA-locus and is coordinated with the strictly conserved E373 at a distance of ~3.3 Å (Fig. 3c, d). We also noticed that

D370/E936 of the DEKA-locus contribute negligibly to the Na2 at distances of 5.6–6.6 Å (Fig. 3c). Thus, we hypothesize that the Na2 may represent the second step for sodium conductance, that is, after captured and partially dehydrated in Na1 site, at least partially-dehydrated Na$^+$ can fit into the Na2 site which is going to enter the narrowest asymmetric constriction site of the SF. The possible partial dehydration of Na$^+$ in the Na2 site is reflected by its relatively weaker density compared to the Na1 (Fig. 3d, e). Furthermore, the K1413 points its long side-chain deep into the SF, forming the narrowest part of the SF. It has been proposed that this residue serves as a key coordination ligand in favor of Na$^+$ or Li$^+$ but is unfavorable for other cations[43]. In line with this hypothesis, Na$^+$ from the Na2 site can quickly pass through the SF and enter the central cavity accelerated by the amino group of the K1413. We found additional elongated density below the K1413 at a distance of ~3.5 Å, which may represent a third Na$^+$ site (namely Na3) (Fig. 3d, e). Consistently, previous MD simulations studies suggested that two Na$^+$ ions spontaneously occupy the symmetric SF of the bacterial Na$_V$ channels, and three Na$^+$ sites were proposed in the asymmetric SF of the eukaryotic Na$_V$ channel[70–72], which are similar to the Na2, Na3 sites and Na1-3 sites of our Na$_V$1.6 structure, respectively.

In Ca$_V$ channels, the Ca$^{2+}$ binding sites were revealed in the SFs[66–68,73], suggesting a possible step-wise "knock-off" mechanism for

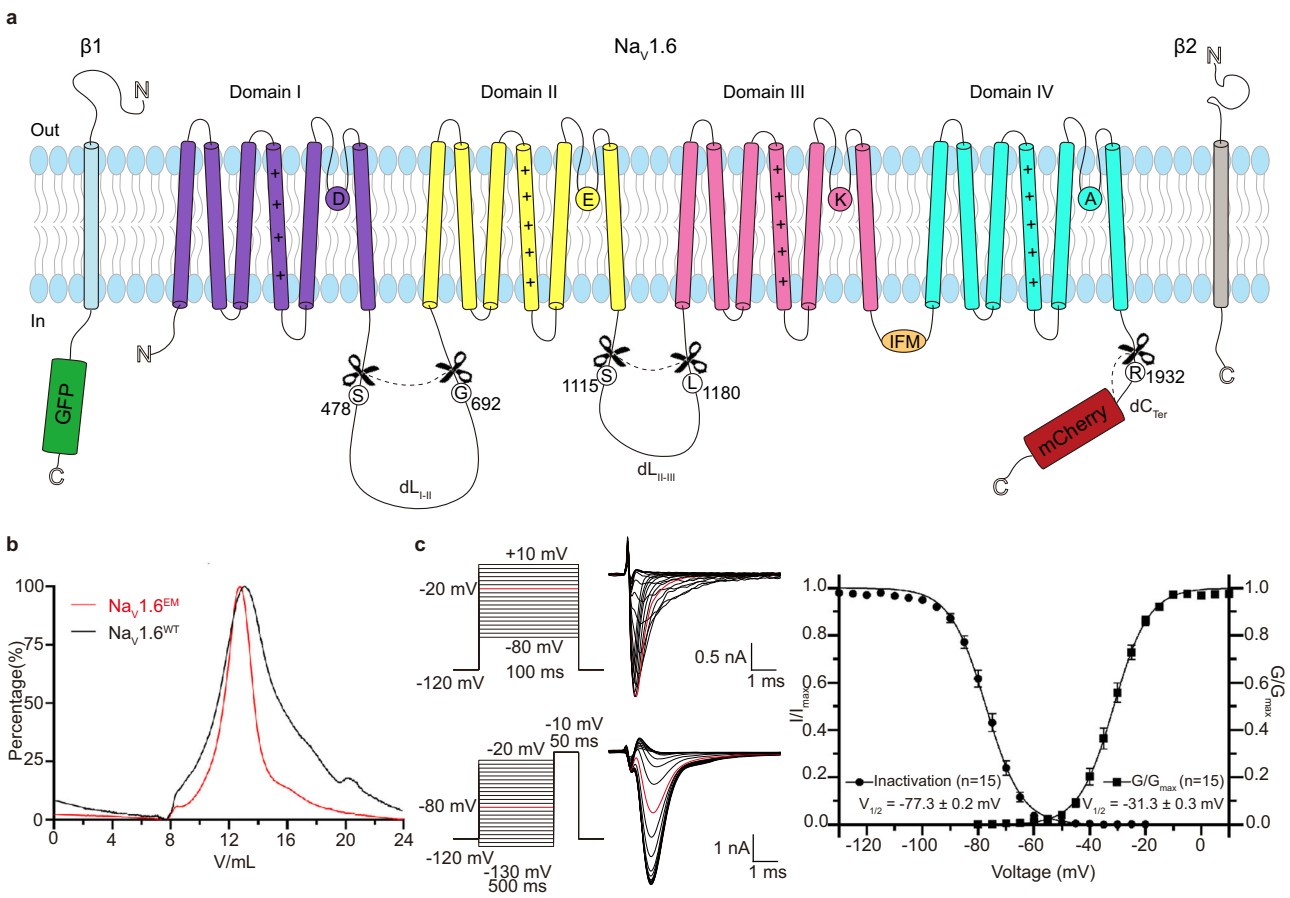

**Fig. 1 | Topology and functional characterization of the Na$_V$1.6$^{EM}$/β1/β2 complex.** a Topology of the Na$_V$1.6/β1/β2 complex. The α subunit consists of DI (purple), DII (yellow), DIII (pink), and DIV (cyan) connected by intracellular linkers, a mCherry fluorescent protein tag fused at the C-terminus. Scissors indicate the truncated sites. The β1 fused with a GFP tag at the C-terminus and the β2 subunit are highlighted in light blue and gray, respectively. The same color codes for Na$_V$1.6/β1/β2 are applied throughout the manuscript unless specified. b Size exclusion chromatogram profiles of the purified Na$_V$1.6$^{WT}$ (black) and the Na$_V$1.6$^{EM}$ (red). c Electrophysiological characterization of the Na$_V$1.6$^{EM}$ construct. The voltage protocols and representative current traces are shown on the left panels. To

characterize the voltage-dependence of activation, Na$_V$1.6$^{EM}$ expressing HEK293T cells were stimulated by a 100 ms test pulse varying from −80 mV to 10 mV in 5 mV increments from a holding potential of −120 mV, with a stimulus frequency of 0.2 Hz. To measure the steady-state fast inactivation, HEK293T cells were stimulated by a test step to −10 mV after a 500 ms prepulse varying from −130 mV to −20 mV in 5 mV increments, from a holding potential of −120 mV and a stimulus frequency of 0.2 Hz. The resulting normalized conductance-voltage (G/V) relationship (squares, $n = 15$) and steady-state fast inactivation (circles, $n = 15$) curves are shown on the right panel. Data are presented as mean ± SEM. $n$ biological independent cells. Source data are provided as a Source data file.

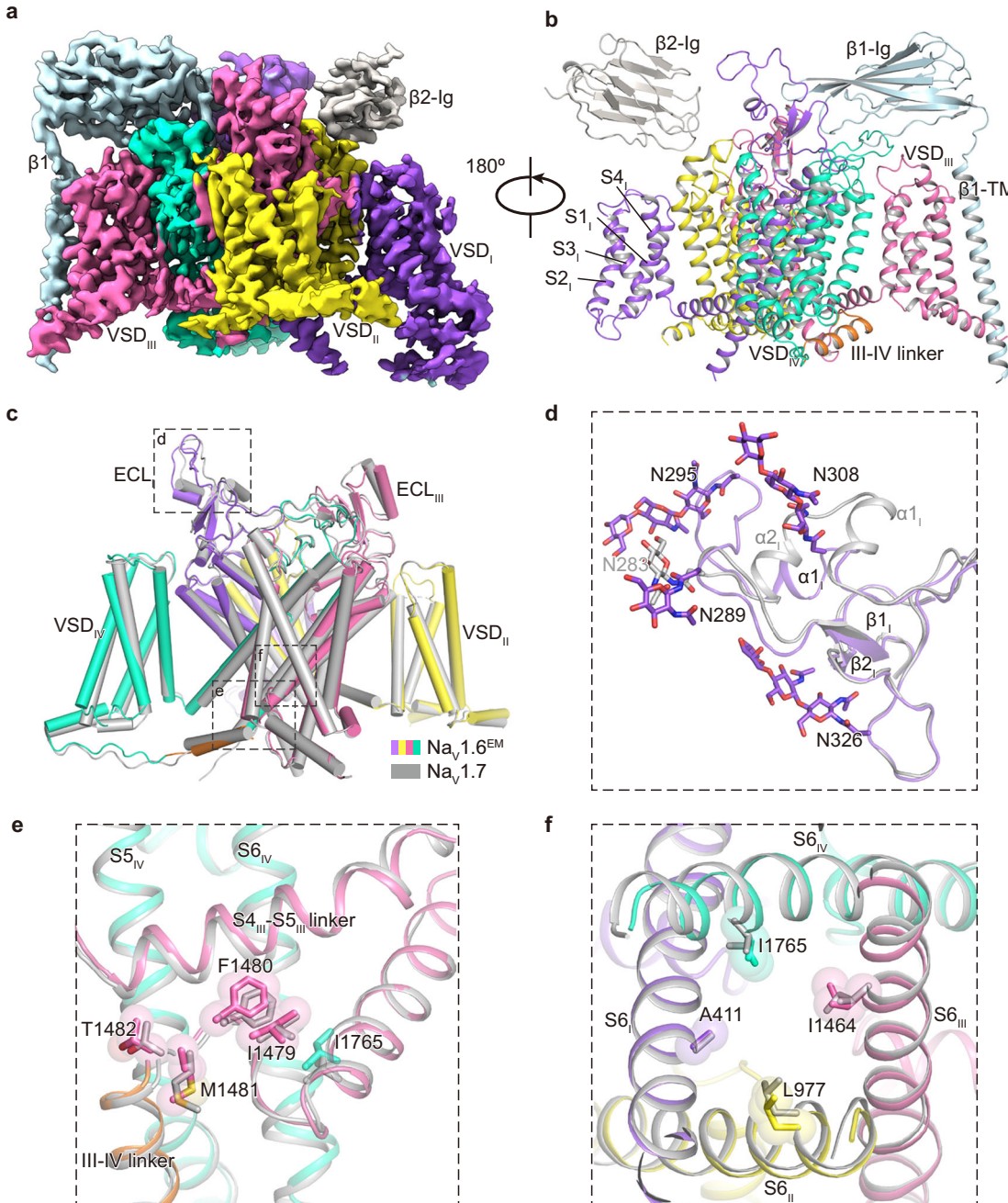

**Fig. 2 | Cryo-EM structure of the Na$_V$1.6$^{EM}$/β1/β2 complex. a, b** The cryo-EM density map (**a**) and cartoon representation (**b**) of the Na$_V$1.6$^{EM}$/β1/β2 complex. **c** Structural comparison of Na$_V$1.6$^{EM}$ and Na$_V$1.7 (PDB code: 7W9K, colored in gray). The black dashed-line squares indicate the areas shown in panels (**d**), (**e**), and (**f**). **d** Superimposition of the ECL$_I$ between Na$_V$1.6$^{EM}$ and Na$_V$1.7. N-linked glycosylation moieties are shown in sticks. **e** Comparison of the IFM motif. The IFM motif were depicted side chains in sticks and spheres with half transparency. **f** Comparison of the intracellular activation gate of Na$_V$1.6$^{EM}$ and Na$_V$1.7 viewed from intracellular side. Key residues from four S6 helices were shown side-chains sticks and spheres with half transparency.

Ca$^{2+}$ conducntance[66]. Superposition of the SFs of the Na$_V$1.6$^{EM}$ and the Ca$_V$Ab shows that the Na1 and Na2 sites are roughly at the same height levels as Ca1 and Ca2 sites in Ca$_V$Ab, respectively (Fig. 3d, e). However, the two Na$^+$ sites are off the central axis of the SF, while the Ca$^{2+}$ sites are in the center (Supplementary Fig. 7). This difference is in agreement with the asymmetric characteristics of the SFs of mammalian Na$_V$ channels. As shown in the Na$_V$1.6$^{EM}$ structure, similar to the Ca$_V$ channels, two or more potential Na$^+$ sites exist in the SFs of Na$_V$ channels. In fact, the SFs of Na$_V$ and Ca$_V$ channels are closely related, point-mutations in the SF of the Na$_V$ channel can convert it into a highly Ca$^{2+}$ favorable channel[66,74]. Nevertheless, these subtle compositional and

conformational differences at the SFs determine the ion selectivity and conductance.

## Blockade of Na$_V$1.6 by 4,9-ah-TTX

The guanidinium neurotoxin TTX and its derivatives can potently inhibit eukaryotic Na$_V$ channels[75]. TTX was reported to be more potent in inhibiting Na$_V$1.6 than other TTX-sensitive Na$_V$ channels[76]. Interestingly, one of the TTX metabolites, 4,9-ah-TTX, has been reported to preferentially block Na$_V$1.6 over the other eight Na$_V$ channel subtypes[54]. We first examined the TTX sensitivity of Na$_V$1.6$^{EM}$, Na$_V$1.2, and Na$_V$1.7, yielding IC$_{50}$ values of 1.9 nM (n = 5), 4.9 nM (n = 5), and

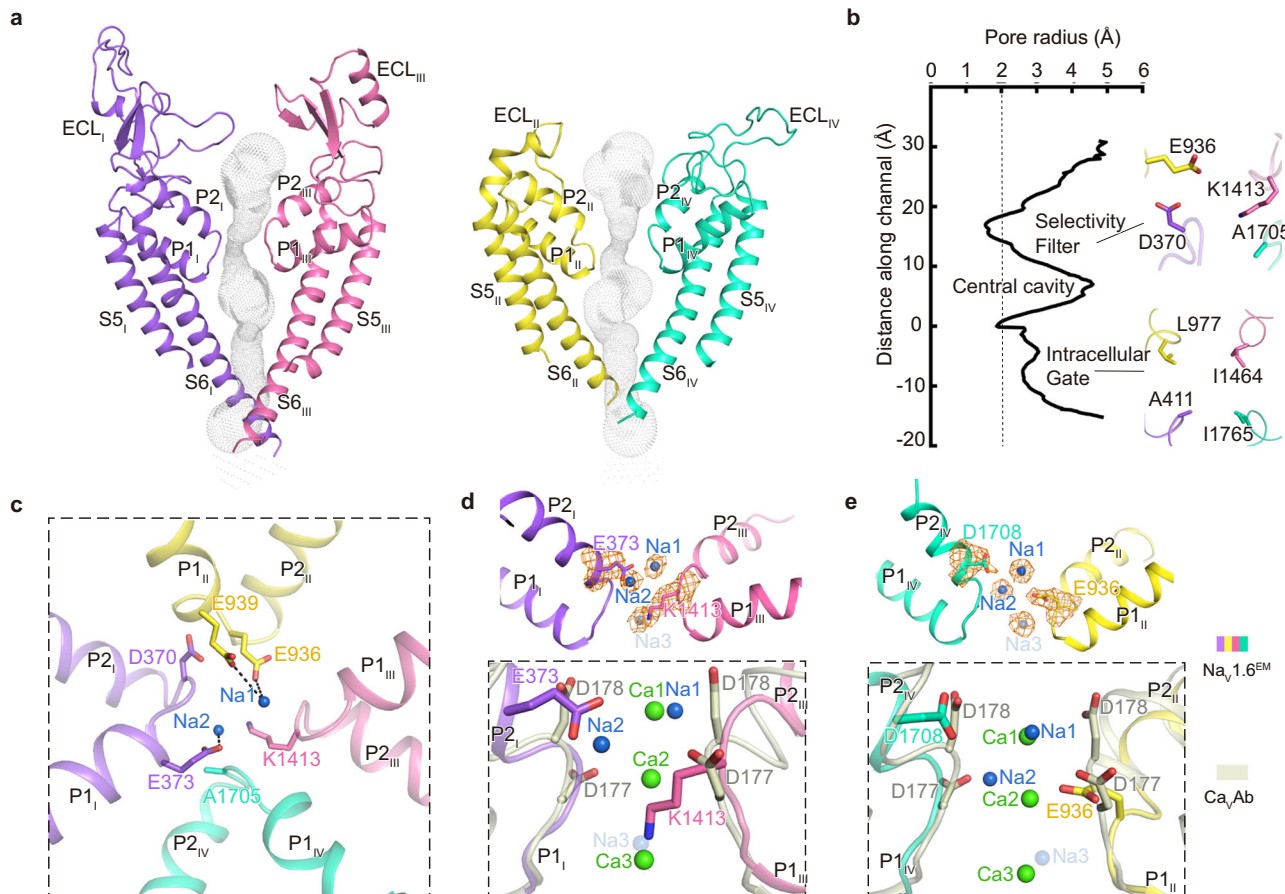

**Fig. 3 | Potential Na⁺ binding sites in the SF of Na_V1.6^EM. a** The ion conductance path of Na_V1.6^EM calculated by HOLE. The diagonal repeats of pore domain only including the S5–S6 and pore-helices were shown for clarity. **b** Plot of the pore radii of Na_V1.6^EM. The dashed line indicates pore radius at 2 Å. The key residues constituting the selectivity filter (SF) and the intracellular activation gate (AG) were shown as sticks. **c** The SF of Na_V1.6^EM viewed from the extracellular side. Potential Na⁺ ions were shown as blue balls. Black dashed lines represent polar interactions. **d, e** Comparison of the Na⁺ binding sites of Na_V1.6^EM and the Ca²⁺ binding sites of Ca_VAb (PDB code: 4MS2, colored in gray). The diagonal repeats of DI and DIII (**d**), DII and DIV (**e**) are shown separately for clarity. The EM densities for putative Na⁺ and key residues are shown in orange meshes contoured at 4 σ and 5 σ, respectively. A third possible Na⁺ ion with weaker density contoured at 3 σ was shown as a light blue ball with half transparency. Ca²⁺ ions are shown as green balls.

16.7 nM ($n = 4$), respectively. Consistent with previous reports, TTX indeed favors Na_V1.6 (Supplementary Fig. 6c). Then we tested the inhibitory effects of 4,9-ah-TTX on Na_V1.2, Na_V1.7, and Na_V1.6^EM. As illustrated in Fig. 4a–c, 4,9-ah-TTX gradually inhibits both Na_V1.7 and Na_V1.6^EM in a concentration-dependent manner. However, the resulting IC_{50} values of 4,9-ah-TTX are significantly different, which are 257.9 nM ($n = 6$) for Na_V1.2, 1340 nM ($n = 6$) for Na_V1.7 and 52.0 nM ($n = 5$) for Na_V1.6^EM, respectively (Fig. 4c). Those results confirmed that the potency of 4,9-ah-TTX is ~27-fold weaker than TTX in inhibiting Na_V1.6^EM, and 4,9-ah-TTX is indeed a Na_V1.6 preferred blocker.

To better understand the underlying mechanism of Na_V1.6 modulation by 4,9-ah-TTX, we solved the cryo-EM structure of Na_V1.6^EM/β1/β2 in complex with 4,9-ah-TTX (named Na_V1.6^{4,9-ahTTX}) at a resolution of 3.3 Å (Supplementary Fig. 4). The overall structure of Na_V1.6^{4,9-ahTTX} is indistinguishable to the Na_V1.6^EM (RMSD at 0.2 Å). However, unambiguous EM density located above the SF of Na_V1.6^{4,9-ahTTX} was observed, which fits a 4,9-ah-TTX molecule very well (Fig. 4d, e, Supplementary Fig. 4b). A closer look shows that the 4,9-ah-TTX occupies the Na⁺ binding sites and sticks into the SF of Na_V1.6 via extensive interactions (Fig. 4f). D370 and E373 from D_I, E936, and E939 from D_II, and D1708 from D_IV form electrostatic interactions with the 4,9-ah-TTX, Y371, and K1413 also contribute to stabilizing the blocker by forming van der Waals interactions (Fig. 4f). Superposition of the

Na_V1.6^{4,9-ahTTX} and the TTX bound Na_V1.7 (Na_V1.7^{TTX}) show a very similar binding mode for the two blockers (Fig. 4f–h). This similar binding mode is reasonable because the chemical structures of TTX and 4,9-ah-TTX are very similar; secondly, these key interacting residues are identical among the TTX-sensitive Na_V channels (Supplementary Fig. 6b). However, subtle conformational differences were observed. The 4,9-ah-TTX binds ~1.4 Å deeper in the pocket of Na_V1.6 than TTX in Na_V1.7 (Fig. 4h). In addition, the 4,9-ah-TTX lacks two hydroxyl groups at the 4 and 9 positions of TTX, which form two more hydrogen-bonds with E364 and G1407 of Na_V1.7, respectively (Fig. 4g). TTX should form the same interactions with Na_V1.6 as found in Na_V1.7. Thus, the binding of TTX to Na_V1.6 is stronger than the binding of 4,9-ah-TTX, which agrees with the higher potency of TTX in inhibiting Na_V1.6 than 4,9-ah-TTX (Fig. 4c and Supplementary Fig. 6c).

Then how does 4,9-ah-TTX preferentially inhibit Na_V1.6 over Na_V1.7 in a nearly identical pocket? By carefully checking the pore-loop sequences of Na_V1.6, we found that L1712 in the D_IV P-loop of Na_V1.6 is a major different residue in the P-loop regions not similar to other Na_V channels (Supplementary Fig. 6b). We tested the effect of 4,9-ah-TTX on L1712A mutant of Na_V1.6 (Na_V1.6^{L1712A}), the resulting IC_{50} value of 4,9-ah-TTX for Na_V1.6^{L1712A} is 61.1 nM ($n = 4$), which is close to that of the Na_V1.6^EM (Supplementary Fig. 6d). This result suggests that L1712 is not relevant to the binding of 4,9-ah-TTX. To test whether the accessibility

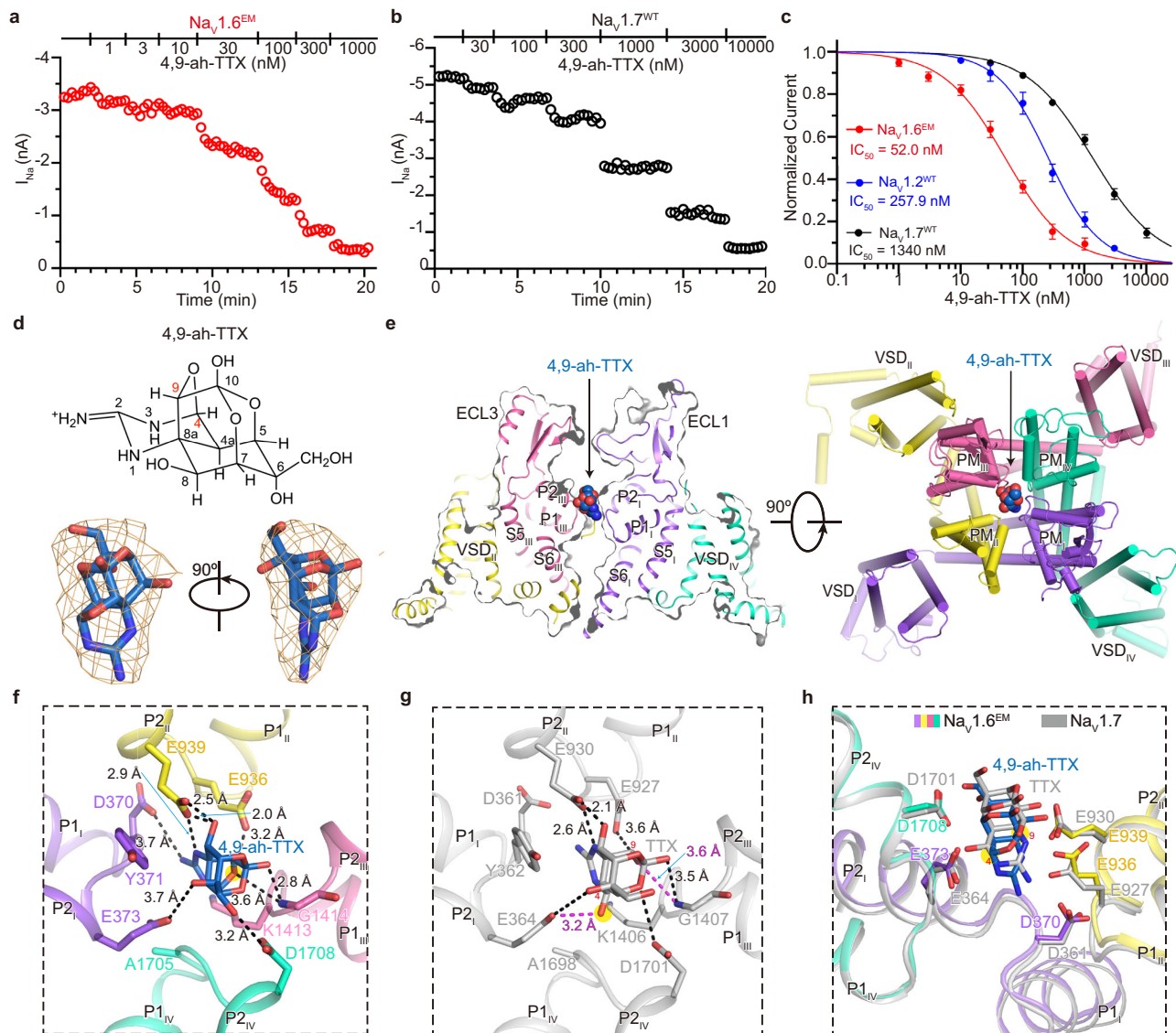

**Fig. 4 | Blockade of the Na$_V$1.6$^{EM}$ by 4,9-ah-TTX. a, b** The peak currents of Na$_V$1.6$^{EM}$ (**a**) and Na$_V$1.7$^{WT}$ (**b**) in response to increasing concentrations of 4,9-ah-TTX. HEK293T cells were held at −120 mV and the inward sodium currents (I$_{Na}$) were elicited by a 50-ms step to −10 mV with a low frequency of 1/15 Hz. **c** The concentration-response curves for the blockade of Na$_V$1.6$^{EM}$ (red), Na$_V$1.2$^{WT}$ (blue), and Na$_V$1.7$^{WT}$ (black) by 4,9-ah-TTX. Na$_V$1.6$^{EM}$, $n$ = 5; Na$_V$1.2$^{WT}$, $n$ = 6; Na$_V$1.7$^{WT}$, $n$ = 6. Data are presented as mean ± SEM. $n$ biological independent cells. **d** The chemical structure of 4,9-ah-TTX (upper panel). The EM density for 4,9-ah-TTX shown in orange meshes contoured at 5 σ (lower panel). **e** The 4,9-ah-TTX binding site in

Na$_V$1.6$^{EM}$. Side (left panel) and top (right panel) view of Na$_V$1.6$^{EM}$ with 4,9-ah-TTX shown in spheres. **f** Detailed interactions between 4,9-ah-TTX and Na$_V$1.6$^{EM}$. Key interacting residues of Na$_V$1.6$^{EM}$ were shown in sticks. Black dashed lines indicate electrostatic interactions between 4,9-ah-TTX and Na$_V$1.6$^{EM}$. **g** Specific interactions between TTX and Na$_V$1.7 (PDB code: 6J8I, colored in gray). The additional hydrogen bonds between Na$_V$1.7 and the 4′, 9′ positions of TTX are highlighted in red. **h** Structural comparison of Na$_V$1.6$^{4,9-ahTTX}$ and Na$_V$1.7$^{TTX}$. The side-chains of key residues in the Na$_V$1.6$^{EM}$ and Na$_V$1.7 depicted in sticks. Source data are provided as a Source data file.

affects the binding of 4,9-ah-TTX, we substituted the ECL$_I$ of Na$_V$1.6 (F273-F356) with that of Na$_V$1.7 (F267-F347) or the ECL$_{III}$ (F1349-V1399) with that of Na$_V$1.7 (F1343-V1392), namely Na$_V$1.6$^{ECL1}$ and Na$_V$1.6$^{ECL3}$, respectively. Surprisingly, the substitution of the ECL$_I$ dramatically drops the IC$_{50}$ values of the 4,9-ah-TTX and TTX by 149-fold and 86-fold, respectively; in contrast, ECL$_{III}$ substitution only decreases the IC$_{50}$ values of the 4,9-ah-TTX and TTX by 2.6-fold and 1.1-fold, respectively (Supplementary Fig. 6d, e). These results show that the ECL substitutions especially ECL$_I$ do affect the potency of TTX analogs, but do not discriminate them.

To further dissect the preferential inhibition of Na$_V$1.6 by 4,9-ah-TTX, we carried out MD simulations of TTX binding to Na$_V$1.6 or Na$_V$1.7, and 4,9-ah-TTX binding to Na$_V$1.6 or Na$_V$1.7. Six independent 100 ns MD simulations were performed for each complex and the

trajectories were used for binding affinity calculations using the method of Molecular Mechanics with Generalized Born and Surface Area solvation (*MM/GBSA*)[77]. The simulation results show that the binding affinity of TTX to Na$_V$1.6 is significantly higher than that of 4,9-ah-TTX to Na$_V$1.6, and the affinity of 4,9-ah-TTX to Na$_V$1.6 is greater than 4,9-ah-TTX to Na$_V$1.7 (Supplementary Fig. 8a). These MD binding affinity results fairly agree with our electro-physiological results (Fig. 4c, Supplementary Fig. 6c). The simula-tions also show that there is only one predominant conformation for 4,9-ah-TTX binding to Na$_V$1.6; while there are four major con-formations for 4,9-ah-TTX binding to Na$_V$1.7 (Fig. 5a, Supplemen-tary Fig. 8b–f). More specifically, E373, E936, and E939 mainly contributed to the binding of 4,9-ah-TTX to Na$_V$1.6, consistent with our structural observation (Fig. 4f, Supplementary Fig. 8f);

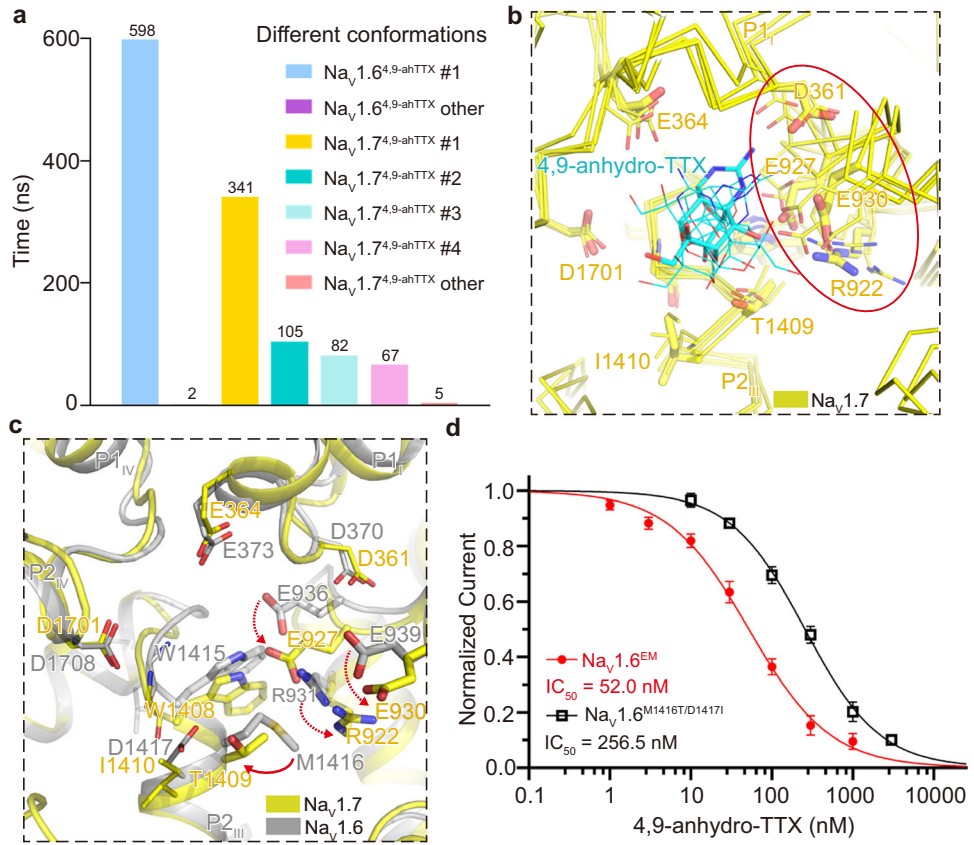

**Fig. 5 | MD simulations of 4,9-ah-TTX binding to Na$_V$1.6 and Na$_V$1.7. a** Cluster analysis of 600 ns molecular simulation trajectories for 4,9-ah-TTX binding with Na$_V$1.6 and Na$_V$1.7, respectively. The clustering was conducted by considering the protein residues within 5 Å of the ligand and using 1.5 Å as R.M.S.D. cutoff. **b** Dynamic behaviors of 4,9-ah-TTX binding in Na$_V$1.7 pocket. Four major conformations of 4,9-ah-TTX bound Na$_V$1.7 were superimposed together, with the most dominant conformation displayed in yellow sticks and other three conformations in wheat lines. The highly flexible region including R922, E927, D361, E930 was indicated by a red circle. The 4,9-ah-TTX was colored in cyan, adopting different poses in the four major conformations. **c** Illustration of the impact of the

small side chain of T1409 to the flexibility of R922, E930, and E927. The red solid-line arrow indicates the size differences between T1409 of Na$_V$1.7 and M1416 of Na$_V$1.6. The gain of the extra flexibility for the side chains of R922, E930, and E927 was indicated by red dashed arrows. Conformation #1 of 4,9-ah-TTX bound Na$_V$1.6 was colored in gray and superimposed with conformation #2 of 4,9-ah-TTX bound Na$_V$1.7 which was colored in yellow. **d** The concentration-response curves for the blockade of Na$_V$1.6$^{EM}$ and Na$_V$1.6$^{M1416T/D1417I}$ by 4,9-ah-TTX. Na$_V$1.6$^{EM}$, $n = 5$; Na$_V$1.6$^{M1416T/D1417I}$, $n = 5$. Data are presented as mean ± SEM. $n$ biological independent cells. Source data are provided as a Source data file.

however, E930 and E927 of Na$_V$1.7, the counterparts of E939 and E936 in Na$_V$1.6, appeared to be very dynamic and contributed less stably to the binding of 4,9-ah-TTX (Fig. 5b). A contact analysis (Supplementary Fig. 9) was conducted to provide more details to understand the dynamics of the ligands (Supplementary Fig. 10). Specifically, E930 and E927 in Na$_V$1.7 interact with 4,9-ah-TTX with a frequency ranging from 21% to 87% for the most populated conformation cluster, whereas the frequency is over 90% for the interactions between such ligand and E939 and E936 in Na$_V$1.6. Superposition of the two representative conformations provides us an assumption that R922 of P1$_{II}$ helix is more flexible in Na$_V$1.7 than the equivalent R931 in Na$_V$1.6 because of the small side-chain T1409 on P2$_{III}$ helix, which in turn increases the flexibility of E930 and E927 and thereby negatively affects the binding of 4,9-ah-TTX to Na$_V$1.7 (Fig. 5c). To validate this assumption, we tested the potency of 4,9-ah-TTX on Na$_V$1.6 with double-mutations of M1416T/D1417I (Na$_V$1.6$^{M1416T/D1417I}$) using whole-cell voltage-clamp recordings. The resulting IC$_{50}$ value is 257 nM ($n = 5$), which is 5-fold less potent than that of Na$_V$1.6$^{EM}$, in coincidence with the findings by MD simulations (Fig. 5d). Taken together, our results confirmed that TTX has the highest affinity to Na$_V$1.6 among the TTX-sensitive Na$_V$ channels; the TTX analog 4,9-ah-TTX is less potent than TTX in inhibiting Na$_V$1.6, but does exhibit preferential inhibition of Na$_V$1.6 over Na$_V$1.7.

## Pathogenic mutation map of Na$_V$1.6

The Na$_V$1.6 channels are abundantly distributed in neurons of both the CNS and the PNS. Compared to other Na$_V$ channel subtypes, the Na$_V$1.6 channel has unique properties including activation at more hyperpolarized potential and generating a large proportion of resurgent current and persistent current, which plays important roles in regulating neuronal excitability and repetitive firing[17,19]. To date, at least 16 gain-of-function mutations in Na$_V$1.6 causing hyperactivity are linked to Developmental and Epileptic Encephalopathy (DEE)[78]; meanwhile, 9 loss-of-function mutations in Na$_V$1.6 causing reduced neuronal excitability are associated with intellectual disability and movement disorders. We highlighted 14 gain-of-function and 7 loss-of-function mutations in our Na$_V$1.6$^{EM}$ structure (Fig. 6). The 14 gain-of-function mutations are mainly distributed in the VSDs, fast inactivation gate, and activation gate. In particular, mutations G1475R, E1483K, M1492V, and A1650V/T target the fast inactivation gate, presumably causing overactivity of the Na$_V$1.6 variants by impairing the binding of the IFM-motif to its receptor site. Mutation N1768D, located at the end of the DIV-S6 helix, was reported to generate elevated persistent current and resurgent current[24,79], which may cause improper gate closing to generate these aberrant currents. Meanwhile, two loss-of-function variants, G964R and E1218K cause intellectual disability without seizure[30]. G964 is located in the middle of S6$_{II}$, which is believed to serve as a hinge in the pore-lining S6 helix during gating[80]. A G964R

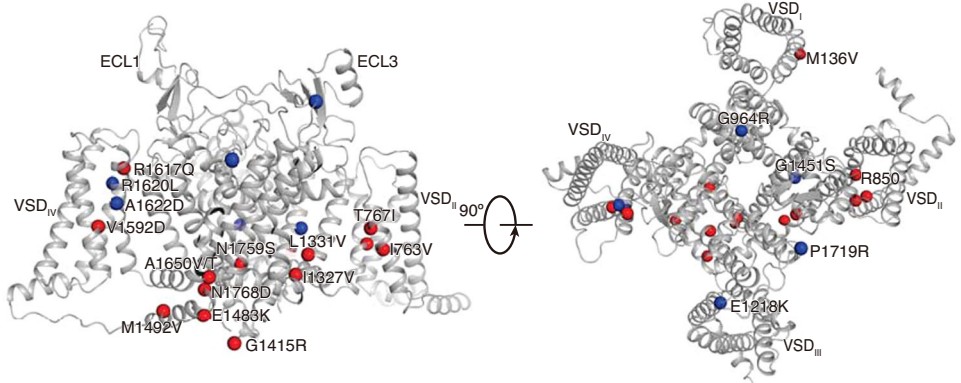

**Fig. 6 | Mapping the pathogenic mutations on the Na$_V$1.6$^{EM}$.** Representative pathogenic mutations were mapped on the Na$_V$1.6 structure. Red and blue spheres represent the gain of function mutations (related to epilepsy) and loss of function mutations (related to intellectual disability), respectively.

mutation can certainly impair the flexibility of the S6$_{II}$ helix; in addition, the additional long side-chain of the mutant can cause clashes with neighboring residues. E1218 belongs to the extracellular negatively-charged clusters (ENCs) of VSD$_{III}$, which play an important role in interacting with the positively-charged gating-charges. The E1218K mutation provides an opposite charge which can disrupt the voltage sensing. This mutant may also destabilize the variant, reflected by its significantly reduced express level[30].

## Discussion

In this study, we presented cryo-EM structures of human Na$_V$1.6/β1/β2 apo-form and complexed with the Na$_V$1.6 preferred blocker 4,9-ah-TTX. To facilitate the structural studies, we obtained the core construct of Na$_V$1.6$^{EM}$ which displayed improved sample quality. This construct and the structures can be a useful tool for future Na$_V$1.6-related structural and biochemical studies. The apo-form Na$_V$1.6 structure reveals three potential Na$^+$ sites, which are coordinated by the important residues in the SF, suggesting a possible mechanism for Na$^+$ recognition, selection, and conductance. By comparison with the Ca$^{2+}$ sites in the bacterial and mammalian Ca$_V$ channels[66–69], the unique asymmetric SF of mammalian Na$_V$ channels provides a precise tunnel to separate Na$^+$ from other cations. However, the exact hydration state of those potential Na$^+$ sites cannot be identified here due to the resolution limit. Future high-resolution structure of Na$_V$1.6 would be required to investigate more detailed mechanisms of Na$^+$ conductance. The 4,9-anhydro-TTX bound Na$_V$1.6 structure demonstrated that 4,9-anhydro-TTX and its closely-related analog TTX share a similar binding pocket, which is composed of nearly identical residues above the SFs. However, TTX has greater potency than 4,9-anhydro-TTX in inhibiting Na$_V$1.6 very likely due to TTX can form two additional hydrogen bonds with Na$_V$1.6. Our MD simulations show that 4,9-anhydro-TTX exhibits a more stable binding mode and greater binding energy with Na$_V$1.6 than Na$_V$1.7. Specifically, the increased flexibility of E930 and E927 may cause the loose binding of 4,9-anhydro-TTX to Na$_V$1.7. Those results potentially explain the higher potency of TTX to Na$_V$1.6 than other TTX-sensitive Na$_V$ channels and the favorable inhibition of Na$_V$1.6 by 4,9-anhydro-TTX. In addition, an interesting observation needed to be mentioned here is the existence of some differences between the binding poses of 4,9-anhydro-TTX in the Na$_V$1.6$^{4,9\text{-}ahTTX}$ EM structure and our MD simulation models (Fig. 4f, Supplementary Fig. 8f). The MD study was conducted with the assumption that the NH group of guanidine in 4,9-anhydro-TTX is fully protonated into NH$_2^+$. However, since such NH in the EM structure is only ~3 Å from the amine group of Y371, it implies an uncertainty of the protonation state of the guanidine of 4,9-anhydro-TTX. When we performed another MD study using unprotonated 4,9-anhydro-TTX and found that the ligand adopts a similar binding pose as observed in

the EM structure. Our findings on the protonation state of 4,9-anhydro-TTX binding with Na$_V$1.6 requires further systemic investigation. Taken together, our results provide important insights into Na$_V$ channel structure, Na$^+$ selectivity and conductance, and modulation by TTX and its analog 4,9-anhydro-TTX.

## Methods

### Whole-cell recordings

HEK293T cells were maintained in Dulbecco's Modified Eagle Medium (DMEM, Gibco, USA) supplemented with 15% Fetal Bovine Serum (FBS, PAN-Biotech, Germany) at 37 °C and 5% CO$_2$. The P2 viruses of Na$_V$1.6$^{EM}$ and Na$_V$1.6 variants were obtained using Sf9 insect cells and used to infect HEK293T cells for 9 h. The plasmids expressing Na$_V$1.2$^{WT}$ or Na$_V$1.7$^{WT}$ were transfected into HEK293T cells using lipofectamine 2000 (Thermo Fisher Scientific, USA). 12–24 h after transfection or infection, whole-cell recordings were obtained using a HEKA EPC-10 patch-clamp amplifier (HEKA Electronic, Germany) and PatchMaster software (HEKA Electronic, Germany). The extracellular recording solution contained (in mM): 140 NaCl, 3 KCl, 1 CaCl$_2$, 1 MgCl$_2$, 10 Glucose, and 10 HEPES (310 mOsm/L, pH 7.30 with NaOH). The recording pipette intracellular solution contained (in mM): 140 CsF, 10 NaCl, 1 EGTA, and 10 HEPES (300 mOsm/L, pH 7.30 with CsOH). The pipettes were fabricated by a DMZ Universal Electrode puller (Zeitz Instruments, Germany) using borosilicate glass, with a resistance of 1.5–2.5 MΩ. The currents were acquired at a 50 kHz sample rate and series resistance (R$_s$) compensation was set to 70–90%. All experiments were performed at room temperature.

Data analyses were performed using Origin 2020b (OriginLab, USA), Excel 2016 (Microsoft, USA), and GraphPad Prism 9.1.1 (Graph-Pad Software, USA). Steady-state fast inactivation (I–V) and conductance-voltage (G–V) relationships were fitted to Boltzmann equations:

$$I/I_{max} = 1/(1 + \exp((V_m - V_{1/2})/k)) \tag{1}$$

$$G/G_{max} = 1/(1 + \exp((V_m - V_{1/2})/k)) \tag{2}$$

$$G = I/(V_m - E_{Na}) \tag{3}$$

where I is the peak current, G is conductance, V$_m$ is the stimulus potential, V$_{1/2}$ is the half-maximal activation potential, E$_{Na}$ is the equilibrium potential, and k is the slope factor.

To assess the potency of 4,9-anhydro-TTX and TTX on Na$_V$ channels, HEK293T cells were held at −120 mV and the inward sodium currents were elicited by a 50-ms step to −10 mV with a low frequency of 1/15 Hz. The concentration-response curves were fitted to a four-

parameter Hill equation with constraints of Bottom = 0 and Top = 1:

$$Y = \text{Bottom} + (\text{Top} - \text{Bottom})/\left(1 + 10^{(X - \lg IC50)}\right) \quad (4)$$

where Y is the value of $I_{Drug}/I_{Control}$, Top is the maximum response, Bottom is the minimum response, X is the lg of drug concentration, and $IC_{50}$ is the drug concentration producing the half-maximum response. The significance of fitted $IC_{50}$ values compared to the control was analyzed using the extra sum-of-squares F test. The drug inhibition statistics are presented in Supplementary Table 1.

## Na$_V$1.6/β1/β2 cloning and expression

The DNA fragments encoding human Na$_V$1.6 (UniProt ID: Q9UQD0), β1 (Uniprot ID: Q07699), and β2 (Uniprot ID: O60939) were amplified from a HEK293 cDNA library. The full-length or truncated Na$_V$1.6, β1, and β2 genes were cloned into the pEG BacMam vector, respectively. For Na$_V$1.6$^{EM}$, residues of inter-domain linkers 478–692, 1115–1180, and 1932 to the last residue were deleted by PCR to optimize the biochemical properties of the purified protein sample. Specifically, Na$_V$1.6$^{EM}$ was fused before a PreScission Protease recognition site, which is succeeded by a mCherry fluorescent protein and a Twin-Strep II tag at the C terminus. A superfolder green fluorescent protein (sfGFP) and His10 tag were introduced at the C terminus of β1. The sequences of all primers used in this study are provided in Supplementary Table 2. For protein expression, recombinant baculoviruses were generated in Sf9 cells using the Bac-to-Bac baculovirus expression system (Invitrogen, 11496015). HEK293F (Gibco, 11625019) cells were cultured under 5% $CO_2$ at 37 °C and were used for transfection at a density of $2.5 \times 10^6$ cells/ml. The Na$_V$1.6$^{EM}$, β1, and β2 viruses were co-transfected into HEK 293F cells at a ratio of 1% (v/v) supplemented with 1% (v/v) FBS. After 8–12 h, sodium butyrate was added into the culture at a final concentration of 10 mM, and the cell was incubated for another 48 h under 30 °C. Cells were then harvested by centrifugation at 1640 × $g$ for 5 min, and finally stored at −80 °C after freezing in liquid nitrogen.

## Purification of human Na$_V$1.6/β1/β2 complex

The Na$_V$1.6/β1/β2 complex was purified following a protocol as was applied in the purification of the Na$_V$1.3/β1/β2 complex[41]. Cells expressing Na$_V$1.6$^{EM}$ complex were resuspended in buffer A (20 mM HEPES pH 7.5, 150 mM NaCl, 2 mM β-mercaptoethanol (β-ME), aprotinin (2 μg/mL), leupeptin (1.4 μg/mL), pepstatin A (0.5 μg/mL)) using a Dounce homogenizer and centrifuged at 100,000 × $g$ for 1 h. After resuspension in buffer B (buffer A supplemented with 1% (w/v) n-Dodecyl-β-D-maltoside (DDM, Anatrace), 0.15% (w/v) cholesteryl hemisuccinate (CHS, Anatrace), 5 mM $MgCl_2$ and 5 mM ATP), the suspension was agitated at 4 °C for 2 h and the insoluble fraction was removed by centrifugation again at 100,000 × $g$ for 1 h. The supernatant containing solubilized Na$_V$1.6$^{EM}$ was then passed through Streptactin Beads (Smart-Lifesciences, China) via gravity flow at 4 °C to enrich the protein complex. The resin was subsequently washed with buffer C (buffer A supplemented with 0.03% (w/v) glycoldiosgenin (GDN, Anatrace)) for 10 column volumes. The purified Na$_V$1.6$^{EM}$ complex was eluted with buffer D (buffer C plus 5 mM desthiobiotin (Sigma, USA)) and was subsequently concentrated to 1 mL using a 100 kDa cut-off Amicon ultra centrifugal filter (Merck Millipore, Germany). The concentrated protein sample was further purified by size exclusion chromatography (SEC) using a Superose 6 Increase 10/300 GL (GE Healthcare) column pre-equilibrated with the buffer E (20 mM HEPES pH 7.5, 150 mM NaCl, 2 mM β-ME, 0.007% GDN). Finally, the fractions containing homogeneous-distributed protein particles were collected and concentrated to ~4 mg/mL for cryo-EM sample preparation.

## Cryo-EM sample preparation and data acquisition

For the preparation of cryo-EM grids, 300-mesh Cu R1.2/1.3 grids (Quantifoil Micro Tools, Germany) were glow-discharged under H2-O2 condition for 60 s. A droplet of 2.5 μL of purified Na$_V$1.6$^{EM}$ complex was applied to the grid followed by blotting for 4−5 s at 4 °C under 100% humidity using a Vitrobot Mark IV (Thermo Fisher Scientific, USA). In the case of the preparation of Na$_V$1.6$^{EM}$ complex with 4,9-anhydro-TTX, 50 μM 4,9-anhydro-TTX (Tocris, UK) was added to the sample before vitrification. Cryo-EM data were collected on a 300-kV Titan Krios transmission electron microscope (Thermo Fisher Scientific, USA) equipped with a Gatan K2 Summit Direct Electron Detector (Gatan, USA) located behind the GIF quantum energy filter (20 e·V). SerialEM[81] was used to collect movie stacks at a magnification of ×130,000 (1.04 Å pixel size) with a nominal defocus range from −1.2 to −2.2 μm. A total dose of 50−60 e$^-$/Å$^2$ was acquired for each movie stack under a dose rate of ~9.2 e-/(Å$^2$s) and dose-fractionated into 32 frames. A total of 3,985 and 2929 movie stacks were collected for the apo- and 4,9-anhydro-TTX-bound Na$_V$1.6 complex, respectively.

## Data processing

For the data processing of apo and 4,9-anhydro-TTX-bound Na$_V$1.6 complex, a similar procedure was performed and a detailed diagram was presented in Supplementary Figs. 3 and 4. All the data were processed in RELION3.0[82] or cryoSPARC[83]. Movies were motion-corrected and dose-weighted using MotionCor2. Contrast transfer function (CTF) estimation was performed with GCTF[84]. Particles were picked using the AutoPick tool in RELION with templates and extracted into 256 × 256-pixel boxes. Several rounds of 2D and 3D classifications were performed to remove junk particles, followed by 3D autorefine, Bayesian polish, and CTF refinement to improve the map quality. The final EM density maps were generated by the non-uniform (NU) refinement in cryoSPARC and reported at 3.4 Å and 3.3 Å, respectively, according to the golden standard Fourier shell correlation (GSFSC) criterion.

## Model building

The sequence of human Na$_V$1.6 and Na$_V$1.7 were aligned using Jalview[85], and a homology model of Na$_V$1.6 was generated using the molecular replacement tool in PHENIX[86]. The atomic models of β1 and β2 subunits were extracted from the structure of Na$_V$1.7 (PDB ID: 6J8I). All of the models were fitted into the cryo-EM map as rigid bodies using the UCSF Chimera[87]. Restraints for 4,9-anhydro-TTX were derived by eLBOW in PHENIX and examined in *Coot*[88]. All residues were manually checked and adjusted to fit the map in *Coot* and were subsequently subjected to rounds of real-space refinement in PHENIX. Model validation was performed using the comprehensive validation (cryo-EM) in PHENIX. The cryo-EM data collection, refinement, and model validation statistics are presented in Supplementary Table 3.

All figures were prepared with UCSF ChimeraX[89] or PyMOL (Schrödinger, USA)[90].

## Molecular dynamics simulations

The structures and force fields for protein, DMPC lipids, and ligands were prepared using the CHARMM-GUI website. The Amber ff14SB force field was used for both protein and lipids with the TIP3P model for water molecules[91]. The GAFF2 force field parameters were used for the ligands[92]. The simulated systems were solvated in water with 150 mM NaCl. The energy minimization was performed using the steepest descent method, followed by six equilibrium steps. During the 2 ns equilibrium steps, the protein backbone atoms were restrained to their initial positions using a harmonic potential with a force constant of 1 kcal mol$^{-1}$ Å$^{-2}$ and the restraints were subsequently removed. Berendsen's coupling scheme was used for both temperature and pressure[93]. Water molecules and all bond lengths to hydrogen atoms were constrained using LINCS[94]. Finally, six independent

production runs were performed for 100 ns. The overall temperature of the system was kept constant, coupling independently for protein, lipids, and solvents at 303.15 K with a Nose-Hoover thermostat[95]. A constant pressure of 1 bar was maintained using a Parrinello–Rahman barostat in a semi-isotropic coupling type for x/y, and z directions, respectively[96]. The temperature and pressure time constants of the coupling were 1 and 5 ps, and the compressibility was $4.5 \times 10^{-5}\,bar^{-1}$ for pressure. The integration of the equations of motion was performed by using a leapfrog algorithm with a time step of 2 fs. Periodic boundary conditions were implemented in all systems. A cutoff of 0.9 nm was implemented for the Lennard–Jones and the direct space part of the Ewald sum for Coulombic interactions. The Fourier space part of the Ewald splitting was computed by using the particle-mesh-Ewald method[97], with a grid length of 0.12 nm on the side and a cubic spline interpolation.

The binding affinities were calculated by MM/GBSA method[98–101]. The MM part consists of the bonded (bond, angle, and dihedral), electrostatic, and van der Waals interactions. The solvation free energies were obtained by using the generalized Born model (GB part), and the non-polar term is obtained from a linear relation to the solvent-accessible surface area (SA part). For each independent trajectory, the first 20 ns trajectory was discarded and 800 frames from 20–100 ns were used for MM/GBSA calculations. The final binding affinity for each ligand-protein complex was obtained by taking the average of the six independent trajectories. Regarding the clustering analysis, structure alignment was first performed for each two of the structures in the trajectory by using Least Squares algorithm which aligns two sets of structure by rotating and translating one of the structures so that the RMSD between matching atoms of the two structures is minimal. Then the clustering analysis was performed by using GROMOS[102] with a RMSD cut-off of 1.5 Å to determine the structurally similar clusters. All the simulations were performed using the GROMACS 2021 suite of programs[103].

### Reporting summary

Further information on research design is available in the Nature Portfolio Reporting Summary linked to this article.

## Data availability

The data that support this study are available from the corresponding authors upon reasonable request. The three-dimensional cryo-EM density maps have been deposited in the Electron Microscopy Data Bank (EMDB) under accession codes EMD-34387 ($Na_V1.6/\beta1/\beta2$) and EMD-34388 ($Na_V1.6/\beta1/\beta2$-4,9-anhydro-TTX). The atomic coordinates have been deposited in the Protein Data Bank (PDB) under accession codes 8GZ1 ($Na_V1.6/\beta1/\beta2$) and 8GZ2 ($Na_V1.6/\beta1/\beta2$-4,9-anhydro-TTX). The UniProt accession codes for the sequences of human $Na_V1.6$, $\beta1$, and $\beta2$ are Q9UQD0, Q07699, and O60939, respectively. The accession codes for the coordinates of $Na_V1.7$, $Ca_VAb$, and $Ca_V3.1$ used in this study are 6J8J ($Na_V1.7$), 4MS2 ($Ca_VAb$), and 6KZO ($Ca_V3.1$). Source data are provided as a Source data file. Source data are provided with this paper.

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

## Acknowledgements

We thank X. Huang, B. Zhu, X. Li, L. Chen, and other staff members at the Center for Biological Imaging (CBI), Core Facilities for Protein Science at the Institute of Biophysics, Chinese Academy of Science (IBP, CAS), and D. Sun at the SM10 Cryo-EM Facility at the Institute of Physics, Chinese Academy of Sciences (IOP, CAS) for the support in cryo-EM data col- lection. We thank Prof. Xuejun Cai Zhang for his helpful discussions, and Yan Wu and Wei Fan for their research assistance service. This work is funded by the Institute of Physics, Chinese Academy of Sciences (E0VK101 and E2V4101 to D.J.), the National Natural Science Foundation of China (T2221001 and 32271272 to D.J., 92157102 to Y.Z., 31871083 and 82271498 to Z.H.), Chinese Academy of Sciences Strategic Priority Research Program (Grant XDB37030304 to Y.Z.), the National Natural Science Foundation of China (Grant 92157102 to Y.Z.), the Chinese National Programs for Brain Science and Brain-like intelligence tech- nology (2021ZD0202102 to Z.H.).

## Author contributions

D.J., Z.H., and Y.Z. conceived and designed the experiments. Y.L. and X.L. prepared samples for the cryo-EM study and made all the con- structs. Y.Q. and B.Y. prepared cells for protein expression. Y.L. col- lected cryo-EM data. Y.L. and D.J. processed the data, and built and refined the models. Y.L. and T.Y. prepared figures. T.Y. collected the electrophysiology data. B.H., F.Z., and C.P. performed MD studies. Y.L., T.Y., B.H., Y.Z., Z.H., and D.J. analyzed and interpreted the results. L.Y., T.Y., B.H., and D.J. wrote the paper, and all authors reviewed and revised the paper.

## Competing interests

The authors declare no competing interests.
