## [Peer Review File · Nature Communications]

Structure of human NaV1.6 channel reveals Na⁺ selectivity and pore blockade by 4,9-anhydro-tetrodotoxinReviewers' Comments:

Reviewer #1:

Remarks to the Author:

Voltage-gated sodium channels play crucial roles in action potential propagation and are important drug targets. Given the high sequence and structural similarities between these channels, it is very difficult to design a drug targeting a specific isoform. In this work, Li et al. solve the structure of human NaV 1.6 w/o a specific pore blocker 4,9-ah-TTX. The structure unambiguously identified the binding mode of the blocker. Further simulation and functional studies suggest the mechanism of the subtype selectivity of 4,9-ah-TTX. The work provides a foundation to further design/optimize blockers targeting NaVs and is valuable to the field.

I have one major concern and a few minor suggestions for the authors.

Major issue:

The sodium binding sites have been observed previously in other NaV structures, such as NaV1.7 at higher resolution (2.2 Å PDB: 7W9K). In this work, the authors observed three densities around the pore and assigned them as sodium ions. Are these sodium ions at the same position as NaV1.7? Is it possible that some of these densities might represent water? Did the authors observe stable sodium binding during MD simulation without TTX? This might be important information for readers to understand the confidence of sodium ion assignment.

Minor issue:

- 1, Line 93: short-form -> shorter-form
- 2, Please show all of the side chain densities of Fig.3 d,e

Reviewer #2:

Remarks to the Author:

The author of this article reports new cryo-EM structures of human NaV1.6 alone and in complex with a guanidinium neurotoxin 4,9-anhydro-tetrodotoxin (4,9-ah- 36 TTX). These structures, in combination with the conduction assays and molecular dynamics simulations, reveal a molecular mechanism of NaV1.6 inhibition by the 4,9-ah- 36 TTX. While the results are novel and well-described, I have a few points and suggestions regarding the lack of thorough analysis of the MD simulation data.

1. In figure 5, the authors have performed clustering analysis on the MD simulation data. It seems like the clustering was done by selecting the protein residues, if that is the case, can the author explain the choice? It makes more sense to do clustering using the ligand atoms or ligand+protein atoms.
2. Also, if the clustering is done on cartesian coordinates, the results become sensitive to the alignment. Please describe how the alignment was performed. Also, describe what clustering algorithm was used.
3. The panel B of figure 3 looks very crowded. It would be better to put the highest populated cluster in the main figure and the rest in the supporting information.
4. Given the fact that the ligand is dynamic in some simulations, please show a root-mean-squared (RMSD) plot of the ligand, separately for each replicate and for all the simulated protein-ligand systems. Also, describe how the alignments were done before RMSD calculations.
5. In supporting figure 8, perform a contact analysis and show the frequency of ligand-protein contact, for each residue separately.
6. In line no. 335, please remove the word "enjoy" and replace it with a more formal and scientifically suitable word.

Response to Reviewers' Comments

We thank the referees for their time to evaluate our manuscript, their overall positive assessment of this study, and their constructive suggestions to improve the manuscript. We have revised the manuscript following the referees' suggestions. A detailed point-by-point response is provided in below.

Reviewers' Comments:

Reviewer #1:

Remarks to the Author:

Voltage-gated sodium channels play crucial roles in action potential propagation and are important drug targets. Given the high sequence and structural similarities between these channels, it is very difficult to design a drug targeting a specific isoform. In this work, Li et al. solve the structure of human Nav1.6 w/o a specific pore blocker 4,9-ah-TTX. The structure unambiguously identified the binding mode of the blocker. Further simulation and functional studies suggest the mechanism of the subtype selectivity of 4,9-ah-TTX. The work provides a foundation to further design/optimize blockers targeting Navs and is valuable to the field.

Reply: We appreciate Reviewer 1's positive comments on the significance and the quality of this study.

I have one major concern and a few minor suggestions for the authors.

Major issue:

The sodium binding sites have been observed previously in other Nav structures, such as Nav1.7 at higher resolution (2.2 Å PDB: 7W9K). In this work, the authors observed three densities around the pore and assigned them as sodium ions. Are these sodium ions at the same position as Nav1.7? Is it possible that some of these densities might represent water? Did the authors observe stable sodium binding during MD simulation without TTX? This might be important information for readers to understand the confidence of sodium ion assignment.

Reply: We thank Reviewer 1's comments for raising this point. We noticed a putative Na⁺ in the high-resolution Nav1.7 structure (PDB code: 7W9K; EMD code: emd_32368). Superposition of the pore-domain of the Nav1.7 and our Nav1.6 shows that the putative Na⁺ ions are located in different positions of the SF (**Response Figure 1a** and **1b**). Surprisingly, the putative Na⁺ of Nav1.7 (orange ball) is out of the ion path, embedded in a cavity formed by D361, E927, and E930 (**Response Figure 1c** and **1d**). In addition, the 2.2 Å resolution Nav1.7 map also exhibits densities in the SF for possible water molecules, the positions of which are not consistent with the putative Na⁺ sites in our Nav1.6 neither; moreover, those densities became almost invisible when the map was low-passed to 3.4 Å (**Response Figure 1e** and **1f**). This observation also suggests that the strong EM densities found in our

Nav1.6 structure at a similar resolution of 3.4 Å is very unlikely for water molecules. The putative Na⁺ sites in our Nav1.6 were modeled based on the strong EM densities and the coordination with the key acidic residues in the SF.

Response Figure 1. Superposition of the pore domain of Nav1.6^{EM} and Nav1.7 (7W9K). **a** and **b**, the putative Na⁺ (orange) in the high-resolution Nav1.7 structure is out of the ion conductance path; the three putative Na⁺ observed in our Nav1.6^{EM} are inside the ion path. **c** and **d**, the detailed position of the putative Na⁺ in Nav1.7. **e** and **f**, EM densities of the putative Na⁺ and key surrounding residues in the high-resolution Nav1.7 (emd_32368 at 2.2 Å) and low-passed to 3.4 Å, respectively.

The studies of Na⁺ binding sites by MD simulations had been performed on both the four-fold symmetric bacterial Nav channels and the asymmetric eukaryotic Nav channel, providing MD information that supports the Na sites in our structure. For instance, Carnevale's (DOI: 10.1021/jz2011379) and Guardiani's (PMID: 28024121) studies suggested that two Na⁺ ions spontaneously occupy two specific binding sites of Site-HFS and Site-IN in the symmetric SF of the bacterial Nav channels, which are at similar positions to that of Na2 and Na3 in our Nav1.6 structure. In addition, three Na⁺ sites were proposed by Xia et al (PMID: 23746512) using MD simulations on the mutated SF of NavRh with the asymmetric DEKA locus, the three Na⁺ sites by the MD simulations appear to be consistent with the Na1, Na2, and Na3 of our Nav1.6 structure. Furthermore, the MD simulations study on the

eukaryotic Nav_vPaS with the same DEKA-locus as the mammalian Nav_v channels by Zhang et al (PMID: 29532417), generated the 3D probability density map for the Na⁺ positions in the SF and identified an asymmetric continuous path through the SF. The three putative Na sites of our Nav1.6 structure fall in the highly-probable permeation areas of the 3D probability density map. We have added a paragraph of the MD studies on the Na sites of Nav_v channel in the revised manuscript.

198 ~3.5 Å, which may represent a third Na⁺ site (namely Na3) (Fig. 3d and e). Consistently,
199 previous MD simulations studies suggested that two Na⁺ ions spontaneously occupy the
200 symmetric SF of the bacterial Nav_v channels, and three Na⁺ sites were proposed in the
201 asymmetric SF of the eukaryotic Nav_v channel⁷⁰⁻⁷², which are similar to the Na2, Na3 sites
202 and Na1-3 sites of our Nav1.6 structure respectively.

Minor issue:

1, Line 93: short-form -> shorter-form

Reply: Thank you. We have corrected this in the revision.

2, Please show all of the side chain densities of Fig.3 d,e

Reply: We thank Reviewer 1's suggestion. We have updated a **New Fig. 3d** and **3e** in the revision.

New Fig. 3d and 3e in the revised manuscript.

Reviewer #2 (Remarks to the Author):

The author of this article reports new cryo-EM structures of human Nav1.6 alone and in complex with a guanidinium neurotoxin 4,9-anhydro-tetrodotoxin (4,9-ah- 36 TTX). These structures, in combination with the conduction assays and molecular dynamics simulations, reveal a molecular mechanism of Nav1.6 inhibition by the 4,9-ah- 36 TTX. While the results are novel and well-described, I have a few points and suggestions regarding the lack of thorough analysis of the MD simulation data.

Reply: We appreciate Reviewer 2's positive comments on the significance of this study, and his/her suggestions

on the MD simulations for improving the manuscript.

1. In figure 5, the authors have performed clustering analysis on the Md simulation data. It seems like the clustering was done by selecting the protein residues, if that is the case, can the author explain the choice? It makes more sense to do clustering using the ligand atoms or ligand+protein atoms.

Reply: We thank Reviewer 2 for raising this point. The clustering was performed by using the ligand and protein residues within 5 Å of the ligand. The previous version of the manuscript didn't make this clear. In the revised version, we have updated the wording accordingly in the legend of Figure 5.

“...*The clustering was conducted by considering the ligand and protein residues within 5 Å of the ligand and using 1.5 Å as RMSD cutoff...*”

To further express our agreement on the reviewer's point of clustering with ligand+protein making more sense than with protein only, we conducted the comparison of the clustering with protein only and with ligand+protein as shown in **Response Figure 2** below. Although the profile of the clusters is generally consistent between the two methods, the clustering with ligand+protein considers more information as input and thus provides more detailed results than that of protein only (e.g. more clusters shown in **Response Figure 2b** for Nav1.7^{4,9-ahTTX}).

Response Figure 2 The clustering results performed by only using the protein residues within 5 Å of the ligand is listed in panel a; results from clustering performed by using the ligand and protein residues within 5 Å of the ligand are listed in panel b. c.# means the index of the cluster.

2. Also, if the clustering is done on cartesian coordinates, the results become sensitive to the alignment. Please describe how the alignment was performed. Also, describe what clustering algorithm was used.

Reply: We thank Reviewer 2's suggestion. To provide more information about the alignment and clustering, we have added the following paragraph to the method section in our revised manuscript:

505 of the six independent trajectories. Regarding the clustering analysis, structure alignment
506 was first performed for each two of the structures in the trajectory by using Least
507 Squares algorithm which aligns two sets of structure by rotating and translating one of
508 the structures so that the RMSD between matching atoms of the two structures is
509 minimal. Then the clustering analysis was performed by using GROMOS¹⁰² with a RMSD
510 cut-off of 1.5 Å to determine the structurally similar clusters. All the simulations were

3. The panel B of figure 3 looks very crowded. It would be better to put the highest populated cluster in the main figure and the rest in the supporting information.

Reply: We thank Reviewer 2's suggestion. In the previous version of the manuscript, the four clusters had already been displayed separately in **Supplementary Figure 8**. The reason we put an all-in-one version in the panel **b** of **Figure 5** is to emphasize that the region indicated by red circle is very dynamic. We agree with Reviewer 2 that the figure look crowded. To display this figure more clearly, we have updated a **New Figure 5b** in the revised manuscript by displaying the highest populated cluster within sticks and the less populated clusters within lines.

New Figure 5b. MD simulations of 4,9-ah-TTX binding to Nav1.6 and Nav1.7. **b.** Dynamic behaviors of 4,9-ah-TTX binding in Nav1.7 pocket. Four major conformations of 4,9-ah-TTX bound Nav1.7 were superimposed together, with the most dominant conformation displayed in yellow sticks and other three conformations in yellow lines. The highly flexible region including R922, E927, D361, E930 was indicated by a red circle. The 4,9-ah-TTX was colored in cyan, adopting different poses in the four major conformations.

4. Given the fact that the ligand is dynamic in some simulations, please show a root-mean-squared (RMSD) plot of the ligand, separately for each replicate and for all the simulated protein-ligand systems. Also, describe how the alignments were done before RMSD calculations.

Reply: We thank Reviewer 2's suggestion. To show the dynamics of the ligands for each replicate of the simulations, we have added a **New Supplementary Figure 10** in the revised manuscript. The description regarding how the alignment is done before RMSD calculation has also been included in the figure legend: Before

RMSD calculation, all the structures in each trajectory were aligned with the initial structure of that trajectory by using Least Squares algorithm.

New Supplementary Figure 10. Ligand dynamics in MD study. a-d. The ligand RMSD plots for each replicate of the simulations for Nav1.6^{4,9-ahTTX} (a), Nav1.7^{4,9-ahTTX} (b), Nav1.6^{TTX} (c), and Nav1.7^{TTX} (d). Before RMSD calculation, all the structures in each trajectory were aligned with the initial structure of that trajectory by using Least Squares algorithm.

5. In supporting figure 8, perform a contact analysis and show the frequency of ligand-protein contact, for each residue separately.

Reply: We thank Reviewer 2's suggestion. The results of contact analysis have been added as **New Supplementary Figure. 9** in the revised manuscript. Below paragraph has also been added to the results sector in the revised manuscript.

283 less stably to the binding of 4,9-ah-TTX (Fig. 5b). A contact analysis (Supplementary Fig.
284 9) was conducted to provide more details to understand the dynamics of the ligands
285 (Supplementary Fig. 10). Specifically, E930 and E927 in Nav_v1.7 interact with 4,9-ah-TTX
286 with a frequency ranging from 21% to 87% for the most populated conformation cluster,
287 whereas the frequency is over 90% for the interactions between such ligand and E939
288 and E936 in Nav_v1.6. Superposition of the two representative conformations provides us

Residues interacting with ligands	Na _v 1.6 ^{4,9} -ahTTX		Na _v 1.7 ^{4,9} -ahTTX					Na _v 1.6 ^{TTX}				Na _v 1.7 ^{TTX}			
	c.1	c.2	c.1	c.2	c.3	c.4	c.5	c.1	c.2	c.3	c.4	c.1	c.2	c.3	c.4
	Cluster population (ns)		342	105	82	67	5	555	32	11	2	479	87	30	3
IP [GLU936 (GLU927)]	100%	100%	87%	97%	100%	28%	60%	100%	84%	100%	100%	100%	100%	100%	100%
HA [GLU936 (GLU927)]	100%	100%	71%	98%	80%	22%	80%	92%	9%	91%	100%	100%	100%	100%	100%
HA [GLU373 (GLU364)]	100%	100%	55%	48%	7%	97%	60%	100%	100%	82%	50%	100%	100%	100%	100%
IP [GLU939 (GLU930)]	98%	100%	21%		87%	78%		91%	34%	73%	100%	39%	76%	43%	33%
IP [ASP370 (ASP361)]	97%	100%	53%	1%	91%	100%	20%	100%	100%	100%	100%	100%	100%	100%	100%
HA [ASP370 (ASP361)]	96%	100%	52%	1%	80%	100%		100%	100%	100%	100%	98%	100%	100%	100%
HA [GLU939 (GLU930)]	93%	50%	46%	91%	91%	78%	100%	92%	31%	82%	100%	62%	70%	73%	67%
HD [MET1416 (THR1409)]	92%	100%	92%	54%	100%	97%	80%	99%	100%	100%	100%	100%	100%	100%	100%
HD [ASP1417 (ILE1410)]	79%	100%	30%		60%	60%		25%	100%	18%		65%	18%	70%	100%
HD [GLY1414 (GLY1407)]	75%	100%	92%	5%	98%	100%	20%	41%	100%	55%	100%	100%	100%	100%	100%
HA [ASP1417 (ILE1410)]	73%	50%						20%	100%	9%					
HD [TRP1415 (TRP1408)]	65%	50%	45%	10%	24%	97%	60%	46%	100%	45%		84%	85%	70%	67%
AR [TYR371 (TYR362)]	54%		21%	53%		61%	60%	32%	6%	9%		8%	15%		
HA [ASP1708 (ASP1701)]	51%	100%			9%	99%		52%	100%			37%	54%	30%	
HD [ARG931 (ARG922)]	47%	50%	48%	65%	45%		80%	10%		9%		53%	2%	87%	100%
HY [TYR371 (TYR362)]	39%	50%	58%	82%		30%	60%	37%	9%	18%		23%	24%	7%	
HD [GLY1706 (GLY1699)]	12%		1%					36%	22%	18%		71%	84%	80%	100%
HA [GLY1414 (GLY1407)]	11%	50%	0%					19%	3%	18%					
HA [PHE1412 (PHE1405)]	5%		16%	27%				4%	31%	9%		11%	11%	20%	
HD [GLY1709 (GLY1702)]	5%	50%						23%		9%					
IP [GLU373 (GLU364)]	5%		38%	54%	20%		60%	34%		9%		73%	2%	100%	100%
HY [LYS1413 (LYS1406)]	1%														
HA [GLY1706 (GLY1699)]	1%							5%							
HA [TYR371 (TYR362)]	1%					40%		1%		50%		2%	1%		
HD [TYR371 (TYR362)]	0%		3%					65%	100%	91%	100%	21%	91%	3%	
HD [ASN374 (ASN365)]	0%							4%							
HD [TRP1707 (TRP1700)]	0%														
HA [GLY317 (GLY308)]			1%												
HD [LEU319 (LYS310)]			2%	6%								0%			
HA [GLN369 (GLN360)]			5%			7%	20%	1%	16%			1%	9%		
HA [CYS934 (CYS925)]			1%			1%									
HA [GLY935 (GLY926)]			47%			93%						1%	16%		
HD [LYS1413 (LYS1406)]			1%					10%	100%			19%	57%		
AR [TRP1415 (TRP1408)]			1%	3%											
HA [MET1416 (THR1409)]			78%	6%	100%	97%		0%				99%	100%	100%	100%
IP [ASP1708 (ASP1701)]			13%	1%								0%			
HA [THR314 (TYR305)]				1%											
HD [TRP917 (TRP908)]				4%			20%								
HD [TYR1420 (TYR1413)]				21%			60%								
HD [TRP372 (TRP363)]								5%		9%					
HA [LYS1413 (LYS1406)]								20%		27%					

New Supplementary Figure 9. Ligand-protein contact analysis based on MD study. The type and frequency of interactions between protein and ligand are listed for each conformation cluster of a protein-ligand system. C.# indicates the index of the cluster. The digits associated with green-white color scheme indicate the appearance frequency of the interaction within the cluster. The frequencies of 4,9-ah-TTX interacting with E930/E927 in Na_v1.7 and with E939/E936 in Na_v1.7 are indicated with red boxes. For residues interacting with ligands, the annotation follows the format of “Interaction Type [Residue in Na_v1.6 (Counterpart the residue in Na_v1.7)]”. The interaction identification and interaction type definition follow the study described by Daria et al (10.1063/5.0019088). A general annotation for interaction types is listed below: IP (salt bridges), HY (hydrophobic interactions), HA (hydrogen bond, ligand atom as acceptor), HD (hydrogen bond, ligand atom as donor), AR (aromatic system related stacking).

6. In line no. 335, please remove the word "enjoy" and replace it with a more formal and scientifically suitable word.

Reply: We thank Reviewer 2's suggestion. The word “enjoy” has been changed to “exhibit” in the revised manuscript.

Reviewers' Comments:

Reviewer #1:

Remarks to the Author:

I have no more questions.

Reviewer #2:

Remarks to the Author:

The authors have addressed all my concerns properly.

REVIEWERS' COMMENTS

Reviewer #1 (Remarks to the Author):

I have no more questions.

Reply: We thank Reviewer 1 again for his/her comments and suggestions that had helped us to revise the manuscript.

Reviewer #2 (Remarks to the Author):

The authors have addressed all my concerns properly.

Reply: We greatly appreciate Reviewer 2's comments and suggestions that had contributed to the improvement of the revised manuscript.